# Calculation of Lipophilicity of Organophosphate Pesticides Using Density Functional Theory

**DOI:** 10.3390/membranes12060632

**Published:** 2022-06-19

**Authors:** Kurban E. Magomedov, Ruslan Z. Zeynalov, Sagim I. Suleymanov, Sarizhat D. Tataeva, Viktoriya S. Magomedova

**Affiliations:** 1Research and Education Center “Smart Materials and Biomedical Applications”, Immanuel Kant Baltic Federal University, Aleksandra Nevskogo Str., 14, 236041 Kaliningrad, Russia; 2Department of Analytical and Pharmaceutical Chemistry, Dagestan State University, 367000 Makhachkala, Russia; actron@yandex.ru (R.Z.Z.); s.sagim.i@ya.ru (S.I.S.); anchemist@yandex.ru (S.D.T.); analitik-dgu@mail.ru (V.S.M.); 3Analytical Center for Collective Use, Dagestan Federal Research Center of the Russian Academy of Sciences, 3670001 Makhachkala, Russia

**Keywords:** DFT, lipophilicity, organophosphate pesticides, toxicity, partition coefficient, logP, extraction, membrane

## Abstract

Higher lipophilicity facilitates the passage of a substance across lipid cell membranes, the blood–brain barrier and protein binding, and may also indicate its toxicity. We proposed eight methods for predicting the lipophilicity of the 22 most commonly used organophosphate pesticides. In this work, to determine the lipophilicity and thermodynamic parameters of the solvation of pesticides, we used methods of density functional theory with various basis sets, as well as modern Grimm methods. The prediction models were evaluated and compared against eight performance statistics, as well as time and RAM used in the calculation. The results show that the PBE-SVP method provided the best of the proposed predictive capabilities. In addition, this method consumes relatively less CPU and RAM resources. These methods make it possible to reliably predict the ability of pesticide molecules to penetrate cell membranes and have a negative effect on cells and the organism as a whole.

## 1. Introduction

In the context of a constant increase in the number of new chemicals introduced into agriculture, an urgent problem is studying their in-depth toxicological qualities in order to develop preventive measures that involve the impact on the human body and the quality of the environment. Among the many xenobiotics that can have a harmful effect on the human body and the environment, pesticides occupy a particularly significant place. This is due to the fact that they are highly biologically active substances, deliberately introduced into the environment, able to circulate and accumulate in it, thereby creating conditions for the possibility of contact with them by the general population and representatives of flora and fauna. In this regard, taking into account the duration, laboriousness and high cost of toxicological studies on warm-blooded animals, the search for and development of alternative methods for determining the parameters of toxicity and danger of xenobiotics is an urgent problem all over the world [1,2,3].

Currently, various mechanisms of the impact of pesticides on biological organisms are known. The main target of pesticides is the cell membrane system. These substances can affect both the structure and function of membranes; they can damage membrane systems directly or indirectly. The direct interaction of pesticides with biological membranes is carried out by binding drugs to any components of the membranes, usually membrane proteins or lipids. The indirect effect is manifested in the disruption of the biosynthesis of proteins, lipids or pigments of cell membranes and leads first to changes in the protein–lipid or pigment composition of the membranes, and then to disturbances in their structure and functioning [4]. Among a large number of different types of pesticides, there are groups that are considered dangerous to humans. The main pesticides that are widely used include organophosphates (OPs), organochlorines and carbamates, which have a high affinity for biological membranes [5]. The main mechanism of action of organophosphate and carbamate pesticides is inhibition of the enzyme acetylcholinesterase, resulting in signs and symptoms of excessive cholinergic stimulation. Unlike organophosphorus pesticide poisoning, carbamate pesticide poisoning tends to be shorter in duration. The main mechanism of action of organophosphate and carbamate pesticides is inhibition of the enzyme acetylcholinesterase, resulting in signs and symptoms of excessive cholinergic stimulation. In contrast to carbamate pesticide poisoning, organophosphorus pesticide poisoning tends to be longer [6]. Since the inhibition of tissue acetylcholinesterase, although reversible, however, organophosphorus pesticides are metabolized more slowly. In turn, organochlorine compounds are neurotoxins with high lipophilicity, chemical stability and persistence in the environment with a long half-life [7,8].

The development of toxicity prediction models for optimal pesticide use, pesticide management and exposure is of great help in monitoring and controlling the harmful effects of pesticide overuse [9,10,11]. Toxicity increases with lipophilicity, as generally higher lipophilicity facilitates the passage of the substance across lipid cell membranes, the blood–brain barrier and protein binding [12]. It is also closely associated with the bioaccumulation and transport of compounds in soil, sediment and groundwater [13]. The lipophilic properties of the compound make it possible to predict its fate in living organisms and offer models for the transfer and accumulation of chemicals in the ecosystem [14]. Lipophilicity is also useful as a characterization of chemicals in relation to their optimal properties for specific biological and non-biological applications. Lipophilicity descriptors determine the ability of endo- and xenobiotics to undergo metabolic transformations and their affinity for targets—protein [15,16]. Lipophilicity is a measure of the affinity of a substance for organic phases. Quantitatively, lipophilicity is defined as the decimal logarithm of the partition coefficient of a substance between water and normal octanol [17]:$$LogP_{O/W}=Log\frac{C_{Octanol}}{C_{Water}}$$

In particular, ionic associates and nonionic substances for use in membrane potentiometric electrodes must have a lipophilicity of 7.4 or more, and for measurements in the blood—at least 11 [18]. High lipophilicity ensures the retention of the target substance in the phase of the electrode membrane and, as a result, a long period of its operation [19]. At the same time, high lipophilicity determines how easily this substance will overcome lipid membranes and, being a xenobiotic, will have a negative impact on the biological organism as a whole. Knowledge of partition coefficients and lipophilicity is very important and can be useful in the development of pesticide cleanup green methods using emulsion–liquid membranes [20].

The authors of the study propose a density functional theory with a continuum solvation model SMD to calculate the partition coefficient and determine the lipophilicity of the 22 most commonly used organophosphate pesticides [21] presented in Figure 1. This paper presents studies to identify the relationship between lipophilicity and toxicity. We were interested in the approach of the authors of this article and decided to supplement this computational study with eight more models and compare their predictive capabilities, as well as the time and computational resources spent, in order to better predict the ability of substances to penetrate biological membranes, and as a result, evaluate their harmful effects.

## 2. Computational Details

In this work, to determine the distribution coefficients ($logK_{O/W}$) and thermodynamic solvation parameters ($\Delta G_{Solv.(Oct.)}^{0}$,$\Delta G_{Solv.(Water)}^{0}$) of the 22 most commonly used organic phosphorus-containing pesticides selected in the work [21], density functional theory methods PBE0, PBE [22] and B3LYP [23] with basis sets def2-SVP and def2-TZVP [24,25], as well as modern Grimm methods PBEh-3c [26] and B97-3c [27] were used. All calculations were performed using the Orca 5.0.3 program [28,29,30,31]. The initial model structures of pesticides were generated in Wolfram Mathematica 13.0.1.0 (Wolfram Research, Inc., Champaign, IL, USA) [32]. At the first stage, the geometry of model structures was optimized in a vacuum at the level of the B97-3c theory, which does not require large computing resources of a personal computer. Further, the stability of each model structure was confirmed by the absence of imaginary frequencies in the calculated IR spectra. Further optimization of the geometry and calculation of the thermodynamic parameters of pesticides were performed at the theoretical levels PBEh-3c, PBE0/def2-SVP/def2-TZVP, PBE/def2-SVP/def2-TZVP and B3LYP/def2-SVP/def2-TZVP without symmetry restrictions, taking into account the dispersion correction D4 [33] and taking into account the influence of the solvent medium (water, octanol-1) according to the SMD model [34].

The distribution coefficient was calculated from the relationship between the equilibrium constant and the difference in the Gibbs free energy of solvation of a solute in octanol-1 and water according to the equation:$$LogP_{O/W}=-\frac{\Delta G_{Solv.(Octanol)}^{0}-\Delta G_{Solv.(Water)}^{0}}{2.302585RT}$$

## 3. Results and Discussion

The results of the calculation of lipophilicities by the proposed methods are presented in Table 1.

The linearity between the experimentally obtained logP${}_{Exp.}$ and the theoretically determined partition coefficient logP${}_{Calc.}$ was obtained for all studied pesticides of OPs (Figure 2 and Appendix A).

Methods for calculating distribution coefficients were evaluated by performance indicators according to [2]. The choice of a suitable forecasting method depends on the accuracy of its work, which we considered in terms of forecasting error. The standard estimates of forecast accuracy (MFA) include the following: mean error (ME), mean absolute deviation (MAD), standard error (MSE), mean percentage forecast error (MPE), mean absolute percentage forecast error (MAPE), correlation coefficient (r), the slope of the linear regression line (SLRL) and the Pearson correlation coefficient (PCC). For ME, MAD, MSE, MPE and MAPE, the closer the estimate is to 0, the more adequately the model describes the experimental data. For r, SLRL and PCCB, that method most adequately predicts the value of the one that is closer to 1. Table 2 presents the results of estimating the forecast error by various MFAs compared with the reference experimentally determined logP for each model.

Figure 2 shows a good dependence of the theoretical lipophilicity on the experimental one, which is confirmed by PCC 0.88677, SLRL 1.002 for PBE-SVP.

Table 2 shows that in terms of accuracy ME, MPE and MAPE, the PBE-SVP method is the most adequate for the experimental data; according to MAD, r and PCC it best corresponds to M06L/6-31, according to MSE, respectively, M062X/6-31 and SLRL, and PBE/6-31. Thus, for three out of eight MFA indicators, the PBE-SVP and M06L/6-31 models showed the best predictive abilities, which make it possible to reliably predict the ability of xenobiotic molecules to penetrate cell membranes and then have a negative effect on cells and the body as a whole.

Calculated solvation free energy change of transfer from the gas phase to the water phase ($\Delta G_{Solv.(water)}$/kcal mol${}^{-1}$) and octanol phase ($\Delta G_{Solv.(Octanol)}$/kcal mol${}^{-1}$) under standard state conditions, and corresponding log KOW values of examined OP pesticide using the eight-method level of theory, with experimentally determined log P are presented in Table 3 and in the Appendix A.

An important characteristic of the model is the resources that the workstation consumes for computing. Table 4 shows the average consumption of computing resources for each logKOW calculation model for the 22 considered connections. The data are presented for the Orca 5.0.3 program. installed in the Linux Debian 11 (bullseye) environment.

Thus, the PBE-SVP method takes an average of 13.5 days to calculate per CPU core, but requires 8000–20,000 MB per CPU core. Less RAM per CPU core is required by the PBEh-3c method, although it takes more time—39.5 days per CPU core.

Knowledge of the mechanisms of action of pesticides makes it possible not only to assess the toxic effect on biological organisms, but also to significantly increase the effectiveness of their use, purposefully modify their structure and properties and synthesize new highly effective drugs. Among the various physicochemical properties of pesticides, the determining influence on the nature of the interaction with the lipid layer is exerted, first of all, by lipophilicity.

## 4. Conclusions

The eight models for predicting the lipophilicity of commonly used organophosphorus pesticides, which are based on the density functional theory PBE0, PBE and B3LYP with the basis sets def2-SVP and def2-TZVP and modern Grimm methods PBEh-3c and B97-3c, proposed in the work, showed good results. Among them, the PBE-SVP model provides the best predictive capabilities and the lowest time costs. These models make it possible to reliably predict the ability of pesticide molecules to interact with cell membranes and have a negative effect on cells and the organism as a whole.

It can be seen from the obtained results that the results of calculations of the SMD model by the density functional theory method at the level of the PBE-SVP theory most closely coincide with the experiment; this combination is also optimal in terms of time spent on calculations for all studied pesticide molecules. The SMD model with the density functional method at the level of the PBE-TZVP theory shows only satisfactory results in terms of accuracy. This difference, in our opinion, can be associated with the fact that the PBE-TZVP method is more suitable for explicit solvent models, taking into account the fine energies of the orientations of solvent molecules on the surface of solute molecules. For implicit SMD models, the PBE-SVP method is sufficient.

In our opinion, in the future it will be interesting to study the correlation of the statistical parameters of the error with the value of dipole moments of pesticides. This correlation will possibly show the advantage of certain models for more accurate prediction of pesticide lipophilicity and toxicity.

## Figures and Tables

**Figure 1 membranes-12-00632-f001:**
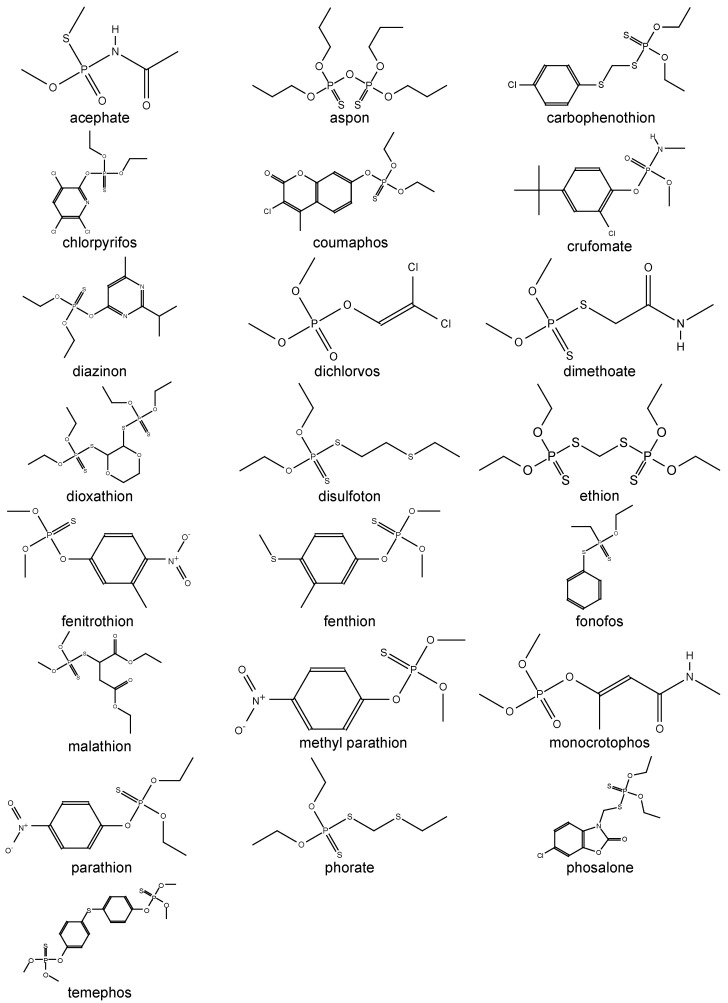
Structure formula OPs.

**Figure 2 membranes-12-00632-f002:**
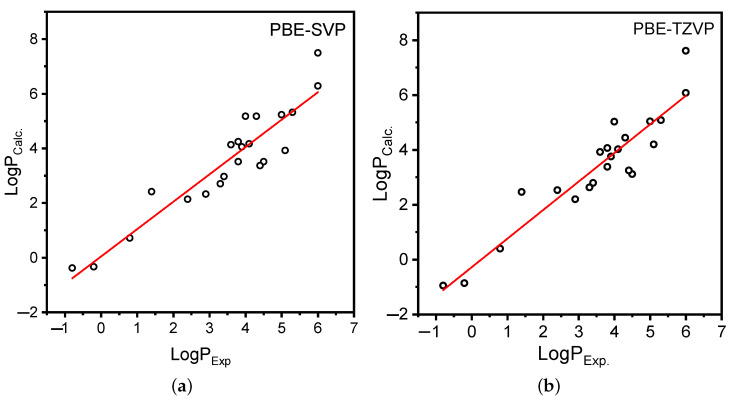
Relation between experimental determined log P and LogP${}_{Calc}$ calculated using PBE with different basis sets—(**a**) SVP and (**b**) TZVP.

**Table 1 membranes-12-00632-t001:** Estimated values of LogP${}_{Calc}$ study of a set of pesticides calculated using various density functional methods with experimentally investigated logP${}_{Exp}$.

No.	Organophosphate	B3LYP-SVP	PBE-SVP	PBE-TZVP	B3LYP-TZVP	PBEh-3c	B97-3c	PBE0-SVP	PBE0-TZVP	LogP${}_{\mathrm{Exp}}$	Ref.
1	Acephate	−0.73	−0.38	−0.95	−0.97	−1.10	−1.10	−0.80	−1.24	−0.80	[21,35]
2	Aspon	7.44	7.49	7.60	7.24	7.19	7.47	7.27	7.30	6.00	[21,35]
3	Carbophenothion	5.23	5.32	5.08	4.90	4.89	4.88	5.05	4.90	5.30	[21,35]
4	Chlorpyrifos	5.45	5.23	5.04	4.61	4.82	4.88	5.28	4.84	5.00	[21,36]
5	Coumaphos	3.30	3.52	3.12	2.88	3.11	2.86	3.25	3.06	4.50	[21,37]
6	Crufomate	2.31	2.97	2.79	2.58	2.42	2.56	2.70	2.60	3.40	[21,35]
7	Diazinon	4.19	4.25	4.06	3.97	3.92	3.90	4.08	4.02	3.80	[21,35]
8	Dichlorvos	2.16	2.41	2.47	2.26	2.07	2.30	2.16	2.33	1.40	[21,35]
9	Dimethoate	0.38	0.71	0.39	0.13	0.10	0.13	0.20	0.12	0.80	[21,35]
10	Dioxathion	5.60	5.18	4.44	5.45	5.32	5.47	5.45	5.57	4.30	[21,35]
11	Disulfoton	4.97	5.17	5.02	4.80	4.78	4.85	4.88	4.97	4.00	[21,35]
12	Ethion	3.54	3.92	4.19	4.11	3.50	4.06	3.21	4.15	5.10	[21,35]
13	Fenitrothion	2.52	2.70	2.63	2.42	2.50	2.51	2.51	2.58	3.30	[21,35]
14	Fenthion	4.16	4.17	4.02	4.13	3.95	3.90	4.07	4.12	4.10	[21,38]
15	Fonofos	3.98	4.06	3.75	3.67	3.63	3.58	3.85	3.65	3.90	[21,35]
16	Malathion	1.89	2.14	2.53	2.22	1.83	2.35	2.23	2.12	2.40	[21,35]
17	Methyl Parathion	2.16	2.32	2.19	2.14	2.15	2.10	2.17	2.32	2.90	[21,35]
18	Monocrotophos	−0.86	−0.34	−0.86	−1.32	−1.36	−1.16	−0.92	−1.22	−0.20	[21,35]
19	Parathion	3.20	3.52	3.38	3.18	3.38	3.13	3.26	3.35	3.80	[21,35]
20	Phorate	4.49	4.13	3.92	4.16	3.97	4.03	4.35	4.29	3.60	[21,35]
21	Phosalone	3.12	3.37	3.25	3.02	2.95	2.92	3.05	3.03	4.40	[21,35]
22	Temephos	5.94	6.28	6.07	5.88	5.76	5.74	5.91	5.91	6.00	[21,35]

**Table 2 membranes-12-00632-t002:** The result of the forecast error estimation for the calculated values of LogP${}_{Calc.}$ (compared to the reference experimentally determined logP) by various MFAs.

MFA	ME	MAD	MSE	MPE	MAPE	r	SLRL	PCC
B3LYP-SVP	−0.12	0.70	0.71	11	35.3	0.9148	1.054	0.8368
PBE-SVP	**0.05**	0.56	0.49	**2.59**	**20.88**	0.9315	1.002	0.8677
PBE-TZVP	−0.13	0.55	0.51	13.45	32.47	0.9356	1.042	0.8754
B3LYP-TZVP	−0.25	0.69	0.66	19.76	46.57	0.9257	1.058	0.8568
PBEh-3c	−0.33	0.71	0.69	19.02	48.36	0.9286	1.064	0.8623
B97-3c	−0.26	0.70	0.70	16.78	43.84	0.9213	1.057	0.8487
PBE0-SVP	−0.17	0.66	0.68	11.19	35.84	0.9164	1.033	0.8397
PBE0-TZVP	−0.19	0.68	0.65	20.79	45.69	0.9288	1.082	0.8627
PBE/6-31 [21]	0.48	0.6	0.58	−5.66	33.83	0.9483	**0.999**	0.8993
M062X/6-31 [21]	0.08	0.45	**0.27**	14.48	28.12	0.962	1.022	0.9255
M06L/6-31 [21]	0.21	**0.44**	0.28	−6.61	24.59	**0.9631**	0.971	**0.9276**

**Table 3 membranes-12-00632-t003:** Calculated solvation free energy change of transfer from the gas phase to the water phase $\Delta G_{Solv.(Oct.)}$/kcal mol${}^{-1}$) and octanol phase ($\Delta G_{Solv.(Water)}$/kcal mol${}^{-1}$) under standard state conditions, and corresponding LogP${}_{Calc.}$ values of examined OP pesticide set at the PBE-SVP level of theory, with experimentally determined log P.

No	Organophosphate	$\Delta G_{\mathrm{Solv}.(\mathrm{Oct}.)}$	$\Delta G_{\mathrm{Solv}.(\mathrm{Water})}$	LogP${}_{\mathrm{Calc}.}$	LogP${}_{\mathrm{Expr}.}$	Ref.
1	Acephate	−60.27	−58.11	−0.38	−0.80	[21,35]
2	Aspon	−18.39	−61.15	7.49	6.00	[21,35]
3	Carbophenothion	−36.73	−67.08	5.32	5.30	[21,35]
4	Chlorpyrifos	−13.10	−42.92	5.23	5.00	[21,36]
5	Coumaphos	−49.89	−69.94	3.52	4.50	[21,37]
6	Crufomate	−44.09	−61.05	2.97	3.40	[21,35]
7	Diazinon	−25.48	−49.70	4.25	3.80	[21,35]
8	Dichlorvos	−16.70	−30.45	2.41	1.40	[21,35]
9	Dimethoate	−60.84	−64.91	0.71	0.80	[21,35]
10	Dioxathion	−47.37	−76.90	5.18	4.30	[21,35]
11	Disulfoton	−26.74	−56.24	5.17	4.00	[21,35]
12	Ethion	−45.09	−67.44	3.92	5.10	[21,35]
13	Fenitrothion	−25.87	−41.28	2.70	3.30	[21,35]
14	Fenthion	−23.46	−47.25	4.17	4.10	[21,38]
15	Fonofos	−37.40	−60.55	4.06	3.90	[21,35]
16	Malathion	−43.06	−55.25	2.14	2.40	[21,35]
17	Methyl Parathion	−25.70	−38.94	2.32	2.90	[21,35]
18	Monocrotophos	−62.72	−60.78	−0.34	−0.20	[21,35]
19	Parathion	−25.55	−45.61	3.52	3.80	[21,35]
20	Phorate	−26.15	−49.70	4.13	3.60	[21,35]
21	Phosalone	−48.89	−68.13	3.37	4.40	[21,35]
22	Temephos	−34.90	−70.76	6.28	6.00	[21,35]

**Table 4 membranes-12-00632-t004:** Average values of computing resources for LogP calculation models.

Average Computing Resources	B3LYP-SVP	PBE-SVP	PBE-TZVP	B3LYP-TZVP	PBEh-3c	B97-3c	PBE0-SVP	PBE0-TZVP	PBE /6-31	M062X /6-31	M06L /6-31
Days per Core CPU	51	13.5	23	126	39.5	21.2	37	133	-	-	-
Memory per Core CPU, Mb	<8000	<20,000	<20,000	<8000	<4000	<4000	<8000	<8000	-	-	-

## Data Availability

No data availability.

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
