# Peer review of "Calculation of Lipophilicity of Organophosphate Pesticides Using Density Functional Theory"

_membranes, 2022, doi:10.3390/membranes12060632_

Round 1
Reviewer 1 Report
The authors developed 8 models for predicting the lipophilicity of commonly used organophosphorus pesticides, which are based on the density functional theory PBE0, PBE and B3LYP with the functional basis sets def2-SVP and def2-TZVP and modern Grimm methods PBEh-3c and B97-3c.The computational method and model establishment are very reasonable, and the computational results also have reference value.In my opinion, the manuscript can be accepted after some minor issues being addressed. Some specific comments are listed below:
1. The number of paragraphs in the introduction and conclusion section is too large, so it is recommended to merge the paragraphs, which seems more beautiful in the format.
2. In the abstract section, it is suggested to express the calculation time briefly to make the summary more concise.
3. In 41-44 Line part, organophosphorus pesticide should be explained as the main object, focusing on the characteristics of it poisoning, rather than taking it as a reference, to explain the characteristics of the other two pesticide poisoning.
4. The authors should express more personal opinions than just simple summaries.
Author Response
Response to Reviewer 1 Comments
Thank you for such valuable questions and recommendations that allowed us to strengthen this work. Below are the responses to yours comments and recommendations:
Point 1: The number of paragraphs in the introduction and conclusion section is too large, so it is recommended to merge the paragraphs, which seems more beautiful in the format.
Response 1: We agree with your remark. Corrections have been made to the text of the manuscript (and highlighted in yellow).
Point 2: In the abstract section, it is suggested to express the calculation time briefly to make the summary more concise.
Response 2: We agree with your remark. Corrections have been made to the text of the manuscript (and highlighted in yellow).
Point 3: In 41-44 Line part, organophosphorus pesticide should be explained as the main object, focusing on the characteristics of it poisoning, rather than taking it as a reference, to explain the characteristics of the other two pesticide poisoning.
Response 3: We agree with your remark. Corrections have been made to the text of the manuscript (and highlighted in yellow).
Point 4: The authors should express more personal opinions than just simple summaries.
Response 4: We agree with your remark. Corrections have been made to the text of the manuscript (and highlighted in yellow).

Reviewer 2 Report
The present manuscript aims to find lipophilicity of some organophosphate pesticides, with the help of theoretical techniques. They have employed continuum solvation model to simulate solvent environment and estimate distribution coefficients and compare with experimental results. Also, they have shown that some methods (which takes dispersion effects into account) provide satisfactory results. Calculations and employed models are quite routine and I could find no interesting results or conclusion. Performance and applications of DFT methods are well understood and conclusions about these methods are trivial. Therefore, I do not recommend publication of this manuscript.
Other points:
1- The authors have stated that initial structures were chosen with respect to symmetry criteria. Most of the considered molecules possess no symmetry. What symmetry criteria are chosen? Moreover, symmetry-based sampling does not necessarily give rise to most minimum structure. Alternatively, the authors should at least (but not limited to) perform a conformational search.
2- The authors have mixed different definitions up, rising phrases with no physical meaning:
What is “functional basis sets”? functional and basis set are two distinct mathematical objects.
What is “ab-initio density functional theory methods”? None of DFTs in this manuscript are ab-initio.
Absence of imaginary frequencies in the calculated IR spectra? Can an imaginary frequency exist in spectrum? Do the authors mean imaginary frequency in harmonic vibrational frequency analysis?
Author Response
Response to Reviewer 2 Comments
Thank you for such valuable questions and recommendations that allowed us to strengthen this work. Below are the responses to yours comments and recommendations:
Point: The present manuscript aims to find lipophilicity of some organophosphate pesticides, with the help of theoretical techniques. They have employed continuum solvation model to simulate solvent environment and estimate distribution coefficients and compare with experimental results. Also, they have shown that some methods (which takes dispersion effects into account) provide satisfactory results. Calculations and employed models are quite routine and I could find no interesting results or conclusion. Performance and applications of DFT methods are well understood and conclusions about these methods are trivial. Therefore, I do not recommend publication of this manuscript.
Response: The methods considered in the article were not previously applied to the determination of the lipophilicity of pesticides and there was no assessment of their accuracy and characteristics of the computing resources used. Such works may be of great interest to the agro-industry and ecology for predicting the properties of new substances. The conclusions are supplemented and highlighted in yellow.
Other points:
1- The authors have stated that initial structures were chosen with respect to symmetry criteria. Most of the considered molecules possess no symmetry. What symmetry criteria are chosen? Moreover, symmetry-based sampling does not necessarily give rise to most minimum structure. Alternatively, the authors should at least (but not limited to) perform a conformational search.
Response 1:
There was a typo in the text. Geometric optimization of molecules was carried out without symmetry restrictions. Conformational search was not carried out, since the molecules are relatively small. The corresponding changes have been made to the manuscript of the article.
2- The authors have mixed different definitions up, rising phrases with no physical meaning:
What is “functional basis sets”? functional and basis set are two distinct mathematical objects.
What is “ab-initio density functional theory methods”? None of DFTs in this manuscript are ab-initio.
Response 2:
We agree with your remark. Corrections have been made to the text of the manuscript (and highlighted in yellow).
Absence of imaginary frequencies in the calculated IR spectra? Can an imaginary frequency exist in spectrum? Do the authors mean imaginary frequency in harmonic vibrational frequency analysis?
Response 3: An imaginary frequency cannot exist in the real experimental spectrum. In the calculated IR spectrum, there can be any imaginary frequency, even a very small one. This shows that the structure is incorrect, and its total energy and other properties based on energy and geometry will be incorrect.

Round 2
Reviewer 2 Report
1- Assessment of DFT methods requires systematic analysis.
2- Neither experimental nor theoretical spectrum, does not include imaginary frequency. On the other hand, vibrational frequency analysis may reveal imaginary frequencies which never appear in the spectrum.